# Wavevector spectral signature of decay instability in space plasmas

Horia Comişel[1,2], Yasuhito Narita[3], and Uwe Motschmann[1,4]

[1]Institut für Theoretische Physik, Technische Universität Braunschweig, Mendelssohnstr. 3, D-38106 Braunschweig, Germany
[2]Institute for Space Sciences, Atomiştilor 409, P.O. Box MG-23, Bucharest-Măgurele, RO-077125, Romania
[3]Space Research Institute, Austrian Academy of Sciences, Schmiedlstr. 6, A-8042 Graz, Austria
[4]Deutsches Zentrum für Luft- und Raumfahrt, Institut für Planetenforschung, Rutherfordstr. 2, D-12489 Berlin, Germany

**Correspondence:** Horia Comişel
(h.comisel@tu-braunschweig.de)

**Abstract.** Identification of a large-amplitude Alfvén wave decaying into a pair of ion-acoustic and daughter Alfvén waves is one of the major goals in the observational studies of space plasma nonlinearity. In this study, the decay instability is analytically evaluated in the 2-D wavenumber domain spanning the parallel and perpendicular directions to the mean magnetic field. The growth-rate determination of the density perturbations is based on the Hall-MHD wave-wave coupling theory for circularly-polarized Alfvén waves. The diagrams of the growth rates versus the wavenumber and propagation angle derived in analytical studies are replaced by 2-D wavenumber distributions and compared with the corresponding wavevector spectrum of density and magnetic field fluctuations. The actual study reveals a perpendicular-shape spectral pattern consistent with the result of a previous study based on 3-D hybrid numerical simulations. The wavevector signature of the decay instability observed in the two-dimensional wavenumber domain ceases at values of plasma beta larger than $\beta$=0.1. Growth-rate maps serve as a useful tool for predictions of the wavevector spectrum of density or magnetic field fluctuations in various scenarios for the wave-wave coupling processes developing at different stages in space plasma turbulence.

## 1   Introduction

Parametric instabilities driven by large-amplitude Alfvén waves have extensively been investigated by analytical studies or numerical simulations in one- or multi-dimensional approaches. A systematic analytical analysis of the multidimensional features of the parametric instabilities have been initiated by Viñas and Goldstein (1991a, b) by applying the Hall-magnetohydrodynamic (hereafter, Hall-MHD) theory to a large-amplitude field-aligned Alfvén wave with left-hand and right-hand circular polarization. Results of the two-dimensional predictions of Viñas and Goldstein (1991a, b) have successfully been confirmed by later numerical simulations. Obliquely propagating daughter waves excited by the decay of a field-aligned Alfvén wave have been observed in 2-D MHD numerical simulations by Ghosh et al. (1993) for a low-beta regime. Other studies on the nonlinear interaction of obliquely-propagating Alfvén waves confirm that the growth rate of the decay instability in direction oblique to the mean magnetic field is typically smaller than the field-aligned decay, see e.g., Mjølhus and Hada (1990), Laveder et al. (2002), Nariyuki et al. (2008). By using two-dimensional hybrid simulations, Matteini et al. (2010) discovered that a broad spectrum of Alfvén and density fluctuations is developing perpendicular to the direction of the mean magnetic field at the decay of a linear

polarized Alfvén pump wave with oblique direction of propagation in low beta plasmas. Gao et al. (2013) reported by means of 2-D hybrid simulations that a linear polarized Alfvén pump wave with parallel propagation can also generate a perpendicular spectrum of daughter waves. Comişel et al. (2019) observed recently a perpendicular spectrum of daughter waves by using field-aligned Alfvén pump waves with circular left-hand polarization and 3-D hybrid simulations. This result was not predicted by previous 2-D numerical simulations. The three-dimensional setup has been also used by Comişel et al. (2020) for analyzing the evolution of large-amplitude Alfvén waves into the azimuthal (or transversal) plane with respect to the mean magnetic field in low-beta plasmas.

The purpose of this study is to recall the former Hall-MHD analytic approach developed by Viñas and Goldstein (1991a). The analytical predictions show that at very low beta values, the oblique decay of a circularly-polarized Alfvén wave becomes competitive with the field-aligned decay. We are looking whether the solutions of the dispersion equation provided by Viñas and Goldstein (1991a) model can or cannot drive a perpendicular spectrum of daughter waves in accordance with the prediction of the 3-D hybrid simulation. In the two-dimensional analytical analysis, the dispersion equation is typically solved by setting a priori the propagation direction of the daughter wave and the complex solution for the frequency is investigated in the wavenumber domain. Here we solve and display the imaginary part of frequencies (namely, growth rates) of the dispersion equation into a wavevector-spectrum like diagram along the parallel and perpendicular directions to the mean magnetic field. On basis on this study, the developed perpendicular spectrum of daughter waves can be considered as a signature for the decay of a left-handed circularly-polarized Alfvén pump wave in low beta plasmas. This pattern describing the oblique-decay process is vanishing for larger values of plasma beta parameter.

**In the two-dimensional analysis of Viñas and Goldstein (1991a), the authors concluded the decrease of the oblique-decay growth rates into narrow band-profiles (in the wavenumber domain) when moderate oblique propagation angles of the daughter waves are considered. In the actual approach, we explore in more details the growth rates at beta values much smaller than the typical value ($\beta$=0.5) used in the previous study. The obtained solutions of the dispersion equations represented in a parallel- and perpendicular- wavenumber diagram reveal new spectral features of the decay instability in low beta plasmas which have never been pointed out in former analytical investigations. Furthermore, this result strengthens the idea of multi-channel coupling of decay instability as a proof-of-concept proposed by Comişel et al. (2019) in a former study based on numerical 3-D hybrid simulations. By following the Hall MHD formalism of Viñas and Goldstein (1991a), we constructed the $k_{\parallel} - k_{\perp}$ spectrum, and the results are in good agreement between the numerical simulation and the analytic treatment.**

## 2 Method and results

We use the analytical analysis developed by Viñas and Goldstein (1991a, b) based on the two-fluid plasma model together with the generalized Ohm's law. The dispersive effects are driven by the ion inertia and the Hall term. Monochromatic parallel-propagating Alfvén waves are exact solutions of the nonlinear MHD equations describing a plasma system. Starting from this property, the set of equations for the evolution of density, flow velocity, and magnetic field is linearized by using a perturbation

expansion in order to define a linear-mode wave (or eigenmode of the system) around the equilibrium of each of the above mentioned quantities. Each linear mode is specified by **its** frequency and wavevector. The wave-wave coupling of the large-amplitude Alfvén pump wave with a density perturbation of wavevector $k$ and frequency $\omega$ is conducting to side-band daughter waves expressed by the relations, $k^{\pm} = k \pm k_0$, $\omega^{\pm} = \omega \pm \omega_0$, where $k^{\pm}$ and $\omega^{\pm}$ **describe the wavevectors and frequencies of the daughter waves, respectively,** $k_0 = k_0 e_{\parallel}$ **with** $k_0$ **- wavenumber of the Alfvén pump wave,** $e_{\parallel}$ **is the unity vector parallel to the mean magnetic field, and** $\omega_0$ **- frequency of the Alfvén pump wave.** The daughter waves are allowed to propagate parallel and obliquely to the magnetic field. The general dispersion equation is derived in terms of 6x6 matrices and depends on six independent parameters: frequency, wavenumber and angle of propagation of the linear mode, amplitude and wavenumber of the pump wave, and plasma $\beta$ parameter. The frequency and growth rate are normalized according to Viñas and Goldstein (1991a) as $\omega_r = Re(\omega)/k_0 V_{\mathrm{A}}$, $\gamma = Im(\omega)/k_0 V_{\mathrm{A}}$, where $V_{\mathrm{A}}$ is Alfvén speed. The plasma beta is defined as $\beta = V_{\mathrm{s}}^2/V_{\mathrm{A}}^2$ with $V_{\mathrm{s}}$ - the sound speed.

The dispersion equation is implemented and solved by using the Mathematica software. We first investigate the solutions $(\omega_r, \gamma)$ for the decay of a right-handed polarized Alfvén wave with same parameters used by Viñas and Goldstein (1991b). Figure 1 (left panel) reports the growth rates (solid line) and frequencies (gray solid line) obtained for plasma $\beta$=0.5. The amplitude of the Alfvén pump wave is 0.2 (normalized to the background magnetic field) and its wavenumber (normalized in terms of ion inertial length) $k_0 V_{\mathrm{A}}/\Omega_{\mathrm{p}}$ is 0.3, where $\Omega_{\mathrm{p}}$ is the ion-gyrofrequency (for protons). The maximum growth rate for parallel propagation is obtained at $k/k_0 \approx 1.25$. The peaks of the growth rates determined at oblique-propagation angles of 10 deg, 20 deg, and 30 deg are decreasing and slightly moving to lower wavenumbers. Their corresponding frequencies are also reducing at larger propagation angles consistent with Viñas and Goldstein (1991b) result. In the same panel we show the solutions of the dispersion equation obtained at a lower plasma $\beta$ value of 0.02. For a better visualization of both frequencies and growth-rate profiles, the values of the wavenumbers along horizontal axis are represented in a logarithmic scale. At small plasma-beta values, the maximum growth rate determined for parallel propagation is significantly larger and is located at $k/k_0 \approx 1.9$ (see black solid line). At an angle of 30 deg, the growth rate is slightly smaller (dotted line) while at a larger value (40 deg), the solutions are splitted into two peaks (thick diamond symbol). The dominant peak has a maximum close to that one derived at parallel propagation. The oblique decays have growth rates similar with that of the field-aligned decay (see also Fig. 3 in Viñas and Goldstein (1991b) at low $\beta$ values). The result for a left-handed polarized Alfvén pump wave is given in the right panel of Fig. 1 calculated at the same low plasma beta value ($\beta$=0.02). The growth rates are slightly smaller for both parallel- and obliquely- propagating daughter waves than those analyzed by considering a right-handed polarized pump wave. At propagation angle of 30 deg, the splitting of the growth rate into two peaks is more pronounced and its maximum value is shifted to larger wavenumbers. In contrast with the left panel of Fig 1, at larger propagation angles (30, 40 deg), the prominent peak of the growth rate (thick diamond) is located in the right-hand side of the plot at wavenumbers $k/k_0 > 2$.

The growth rates derived from the dispersion equation shown in Fig. 1 can be visualized in a different way by constructing a wavevector diagram or growth-rate map analogue to the representation of the 2-D wavenumber spectrum of density or magnetic field fluctuations. The mapping of the growth rates in such coordinates is helpful in our study for a two-folded purpose discussed bellow for: (i) establishing or finding of a specific pattern for the parametric decay in the wavevector domain

and (ii) direct comparison of the analytical predictions with spectra of fluctuations obtained from numerical simulations (or presumably in-situ measurements).

The top panels of Fig. 2 are the arrangement of the plots shown in Fig. 1 into the parallel and perpendicular wavenumber domain for the right-handed and left-handed Alfvén pump waves. The growth rates in the new coordinates are obtained by solving the dispersion relation for $k/k_0$ wavenumbers spanning the domain (1,3) and (0,3) along the parallel and perpendicular direction with respect to the main magnetic field. **Because this procedure is a demanding numerical task, the solutions of dispersion equations are searched in a limited wavenumber range where decay instability is expected (for instance, the domain below $k_\parallel/k_0 = 1$ where modulational instability could be operational is omitted).** The solution is determined for a given pair of values $(k_\parallel, k_\perp)$ which is then advanced by a discrete $\Delta k$ step for each parallel or perpendicular direction. The resulted solutions are smoothed and represented in the wavevector domain. One may observe an "arc"-shape branch of solutions and a perpendicular one for the right-handed polarized Alfvén pump wave. For left-handed polarized waves, the perpendicular branch clearly dominates the "arc"-shape branch in accordance with the profiles drawn in Fig. 1. The bottom panels of Fig. 2 present the map of growth rates determined at larger beta values for left-handed polarized Alfvén pump waves. While the parallel wavenumber of the maximum growth rate is shifting to lower values, the perpendicular branch becomes weaker and at larger beta values ($\beta > 0.1$) the oblique-decay becomes insignificant with respect to the lowest-beta analyzed case. From this analysis we conclude that a perpendicular spectral pattern of the decay products can be associated with the decay process of a circularly polarized Alfvén wave in low-beta plasma.

In Fig. 3 we compare the result of the actual study for the left-handed polarization and beta value of 0.02 with the 2-D wavenumber spectrum of density and magnetic field fluctuations from a former 3-D hybrid simulation (Comişel et al., 2019). First, the growth-rate map given in Fig. 2 is extended towards both positive and negative perpendicular wavenumbers. Second, the growth rates for the lower side-band daughter waves ($\omega_- = \omega - \omega_0$) describing Alfvén waves with backward propagation are added at (negative) wavenumbers $k_\parallel^- = k_\parallel - k_0$ on the basis that $Im(\omega_-) \equiv Im(\omega)$ according to the wave-wave coupling scheme. The magnetic field and density fluctuations given in right panel of Fig. 3 are obtained from a former 3-D hybrid simulation based on AIKEF code (Müller et al., 2011) and a similar scenario with the current study. **Magnetic field fluctuations are represented at negative wavenumbers corresponding to the backward propagating Alfvén daughter waves while the compressional forward propagating waves (acoustic-like) are shown as density fluctuations along the positive wavenumbers. Furthermore, the 2D wavenumber spectra are filtered in frequency domain such that only the frequencies expected for the Alfvén daughter waves at $\omega = \omega_s - \omega_0$ and for density mode at $\omega = \omega_s$ (here $\omega_s$ is the frequency of the ion acoustic wave) are shown.** The spectral analyses report the Alfvén daughter modes at $k_\parallel V_A/\Omega_\mathrm{p} \sim -0.2$ and the sound daughter waves at $k_\parallel V_A/\Omega_\mathrm{p} \sim 0.4$. There is a good qualitative match between the two panels. As we already mentioned, the analytical model does not consider the wave damping or the harmonics of the excited daughter modes. The actual study and former hybrid simulation suggest that the field-aligned decay is accompanied by an oblique-decay process developing a perpendicular spectrum of density and magnetic field fluctuations.

## 3   Discussion

Former studies based on 2-D hybrid simulations, e.g., Matteini et al. (2010), reported a transversal spectrum of daughter waves observed at the decay of a large amplitude Alfvén wave with linear polarization in low beta plasmas. Matteini et al. (2010) discussed the observed result in terms of the finite oblique angle of propagation of the imposed Alfvén pump wave with respect to the mean magnetic field. Gao et al. (2013) confirmed the former 2-D numerical study and even noticed a similar perpendicular spectrum of waves excited at the decay of a field-aligned pump wave. Two decades before, the two-dimensional analytical studies of Viñas and Goldstein (1991b) suggest, on basis of the beta dependence of the oblique-decay growth rates, that the decay process of a field-aligned circullary-polarized Alfvén wave might become important at finite angles of propagation for the daughter waves in low beta plasmas. The mechanism controlling the oblique decay instabilities has been discussed in relation with the coupling between the electrostatic (dominant at parallel propagation) and electromagnetic (dominant at high oblique angles) terms in the Hall MHD non-linear equations. By representing the growth rates provided by Viñas and Goldstein (1991a) formalism in the wave-vector domain, the perpendicular spectrum of daughter waves can be straightfully noticed in Fig. 2, while this feature is not obvious in the usual representation in Fig. 1. The perpendicular decay of daughter waves have been therefore predicted by the Hall MHD theory well before the first observations provided by 2D hybrid simulations in low beta plasmas. On basis of this result, one may expect that growth-rate maps in the wave vector domain could reveal new properties of parametric instabilities which can be further investigated or diagnosed by analytical treatments or by numerical simulations.

There are still open questions started in early studies about the role played by parametric instabilities to the generation of turbulent cascades, (Hoshino and Goldstein, 1989; Ghosh and Goldstein, 1994). The perpendicular spectrum of daughter waves triggered by the nonlinear wave-wave coupling processes is superposed to the perpendicular spectrum of plasma turbulence generated by the post-saturation processes of the decay instability. The plasma anisotropy index as a quantitative measure of the directional (perpendicular) turbulence evolution describes the wavevector anisotropy of a two-dimensional spectrum of magnetic fluctuations (Shebalin et al. , 1983). An increasing number of 2-D or even 3-D wavevector spectra are observationally being reported from in-situ solar wind measurements, see e.g., Narita et al. (2010); Narita (2014a); Narita et al. (2014b). It would be interesting to compare the beta dependence of the wavevector anisotropy of solar wind turbulence, see e.g., Comişel et al. (2014), with the anisotropy determined from the growth-rate maps of decay instability in its dependence on plasma beta parameter as they are displayed in Fig. 2.

To our knowledge, the current study is the first step to bring together the results of the Hall MHD analytic approach and kinetic simulations, in emphasizing the generation of oblique daughter waves through their nonlinear evolution described by the MHD model. One may also notice that the MHD description can be helpful to clarify observed properties of the parametric instabilities which could originate in the kinetic approach by non-physical fluctuations due to the particle discreteness expected in hybrid or particle-in-cell simulations.

**As limitations of the Viñas and Goldstein (1991a) method used in this analysis, the linear dispersion was restricted to excitation of only fundamental side-band daughter waves (i.e., no harmonics are allowed). The electron-inertia effect is neglected; one may expect that time scales or length of interest are larger than the electron gyroperiod or electron inertial length, respectively. The validity of the MHD approach with included Hall term is thus limited up to higher frequencies close to the ion gyro-frequency. The Landau and cyclotron damping effects have also been neglected in the MHD approach.**

## 4  Summary

In conclusion, the analytic method developed by Viñas and Goldstein (1991a) prescribes that in low beta plasmas, circularly polarized Alfvén waves decay into parallel and obliquely-propagating daughter waves. The growth-rate values of the decay process plotted in the two-dimensional domain of the wavevector parallel and perpendicular to the mean magnetic field evince a displacement of the solutions into two branches: a perpendicular one predominant for both left- and right- hand polarization and an "arc"-shape one which is stronger for the right-hand polarization. The oblique decay significantly decreases at beta values larger than $\beta > 0.1$. The theoretical prediction for the left-handed polarized Alfvén pump wave derived in the 2-D wavenumber domain is consistent with the 2-D spectrum of density and magnetic field fluctuations resulted from 3-D hybrid simulations.

Growth-rate maps as those discussed above can be conveniently obtained for various values of the input parameters describing the dispersion equation. A catalogue of maps realized by a systematic analyzing for plasma beta, amplitude of pump wave, polarization or amount of dispersion can provide valuable information for further investigations of parametric instabilities by using hybrid or full-particle numerical simulations in two- or three- dimensional approaches. The growth-rate maps derived by analytical models and subsequently confirmed by numerical simulations can be helpful in future studies as predictions for the spectrum of density or magnetic field fluctuations expected from in-situ measurements. Furthermore, the particular signature of the oblique decay can serve as an evidence of wave-wave coupling processes acting at different evolution stages in space plasma turbulence.

*Author contributions.*  HC worked on theory implementation and manuscript writing. YN worked on representation of wave nonlinearities and application to space plasmas. UM worked on discussion and supervised of the study. All authors read and approved the final manuscript.

*Competing interests.*  The authors declare that they have no conflict of interest.

*Acknowledgements.*  This work is financially supported by a grant of the Deutsche Forschungsgemainschaft (DFG grant MO539/20-1). The work conducted by HC in Bucharest is supported by ESA project MAGICS, PRODEX contract C4000127660. YN is grateful to Masahiro

Hoshino and staff in his group at the University of Tokyo for their hospitality and discussion during the research visit, which was supported by the Japan Society for the Promotion of Science, Invitational Fellowship for Research in Japan (Long-term) under grant FY2019 L19527.

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

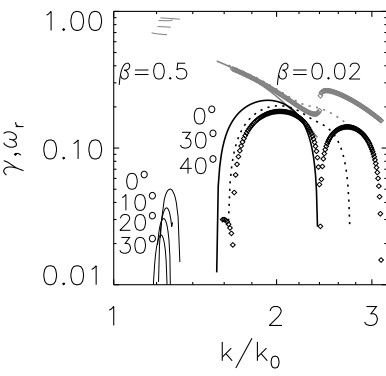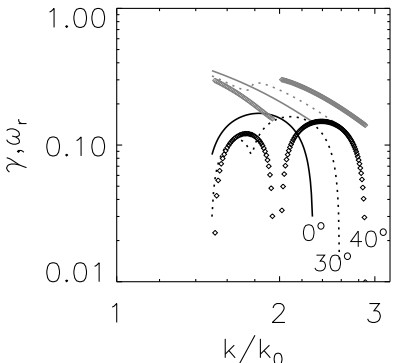

**Figure 1.** Solutions of the dispersion equation (growth rates in black and frequencies in gray) versus normalized wavenumber for right-handed (panel left) and left-handed (panel right) polarized Alfvén pump waves at propagation angles $0$ deg, $30$ deg, and $40$ deg and plasma $\beta$=0.02. Left panel also shows the result obtained for right-hand polarization and plasma $\beta$=0.5.

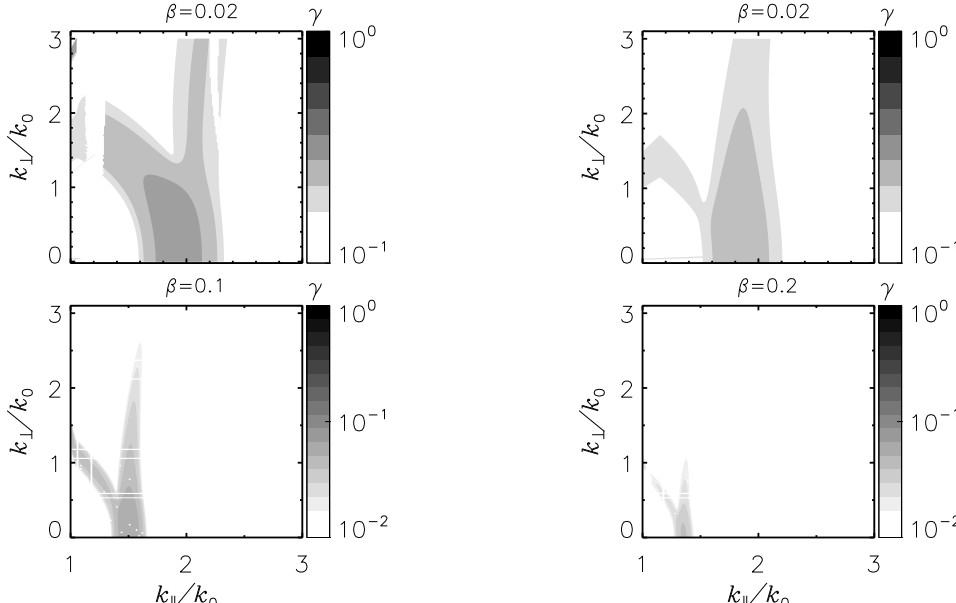

**Figure 2.** Top: Map of the growth rates in the $k_\parallel - k_\perp$ wavenumber domain **for right- (panel left) and left- (panel right) handed polarization of the Alfvén pump wave** corresponding to the analysis shown in Fig. 1 at plasma $\beta$=0.02. Bottom: Diagrams for left-handed polarized Alfvén pump wave at higher beta values.

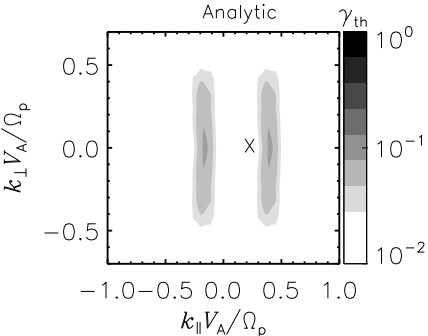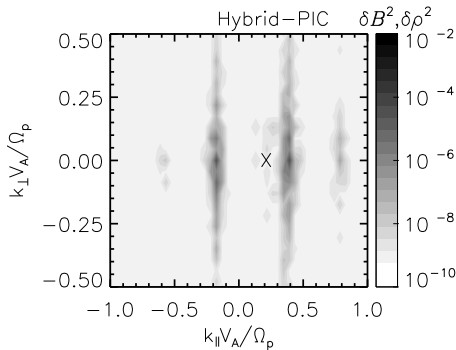

**Figure 3.** Comparison between the extended map of the growth rates in the $k_{\parallel} - k_{\perp}$ wavenumber domain (left panel) and 2-D wavenumber spectrum of density and magnetic field field fluctuations (right panel) from Comişel et al. (2019). **The location of Alfvén pump waves is marked by a cross symbol**.