# Peer review of "Wavevector spectral signature of decay instability in space plasmas"

_Annales Geophysicae, 2020_

## Referee Comment (RC1) · Anonymous Referee #1 · 1 Jul 2020

The manuscript is presenting the results of analytical model describing parametric decay to explain earlier published results of 3D numerical simulations by the same authors. The manuscript requires some improvements before being publishable.

Points to be addressed:

* MAJOR. It is not clearly specified what are the new results in comparison with Viñas and Goldstein (1991) results and other recent studies. The reader should clearly see what has been done until now and what is the new addition to the field. * MAJOR. Discussion section is missing. Currently the manuscript is not clearly demonstrating that the obtained results are important for something. In the abstract there is a statement "Growth-rate maps serve as a useful tool for predictions of the wavevector spectrum of density or magnetic field fluctuations in various scenarios for the wave-wave coupling processes developing at different stages in space plasma turbulence." It would be very useful if authors could discuss at least some scenario showing that their quantitative/qualitative results are applicable and important. Also it would be good to discuss the limitations of the method.

* lines 48-51. The description of k+- and w+- should clearly specify that those describe daughter waves. ko is vector but described in text as field-aligned wavevector, this is confusing. * line 48. What is "state vector", it is never used in the manuscript. * line 53. It should be more clearly explained what authors mean by "limitation for the frequency domain". * line 87. The meaning of "domain (1,3) and (0,3)" not explained. * line 104-105. Needs more clarification how one distinguishes Alfvén and sound daughter modes, as the figure shows only the magnetic field magnitude oscillations. For parallel propagation waves are not compressional so it is surprising to see magnetic field compressions. * Figure 2: caption does not specify the difference between top left and top right panels. * Figure 3: It would be good to mark the location of pump waves booth in analytical and simulation results. right panel) are magnetic field and density fluctuations normalised to something, why are magnetic field and density fluctuations the same? It is not the case in other simulations. * Outlook section should be named "Summary" or "Summary and outlook".

---

## Author Comment (AC1) · 6 Jul 2020

We thank very much the reviewer for reading our manuscript and raising helpful comments and remarks. We will answer to the reviewer's requirements in the upcoming revised manuscript.

―――――――――――――――――――

---

## Author Response (AR1)

**Reply to referee comments**

Wavevector spectral signature of decay instability in space plasmas

H. Comişel, Y. Narita, and U. Motschmann

We thank very much the reviewer for carefully reading our manuscript and for providing valuable suggestions and comments. Here we give our reply. The changes in the manuscript are marked in boldface.                    5

*Reviewer:*
*The manuscript is presenting the results of analytical model describing parametric decay to explain earlier published results of 3D numerical simulations by the same authors. The manuscript requires some improvements before being publishable. Points to be addressed:*                    10

   *\* MAJOR. It is not clearly specified what are the new results in comparison with Viñas and Goldstein (1991) results and other recent studies. The reader should clearly see what has been done until now and what is the new addition to the field.*

Reply:
                                                                                                    15
In the two-dimensional analysis of Viñas and Goldstein 1991, the authors concluded the decrease of the oblique-decay growth rates into narrow band-profiles (in the wavenumber domain) when moderate oblique propagation angles of the daughter waves are considered. In the actual approach, we explore in more details the growth rates at beta values much smaller than the typical value ($\beta$=0.5) used in the previous study. The obtained solutions of the dispersion equations represented in a parallel- and perpendicular- wavenumber diagram reveal new spectral features of the decay instability in low beta plasmas which have never    20
been pointed out in former analytical investigations. Furthermore, this result strengthens the idea of multi-channel coupling of decay instability as a proof-of-concept proposed by Comisel et al. 2019 in a former study based on numerical 3-D hybrid simulations. By following the Hall MHD formalism of Viñas and Goldstein 1991, we constructed the $k_\parallel - k_\perp$ spectrum, and the results are in good agreement between the numerical simulation and the analytic treatment.

Changes in manuscript:                                                                                25
Page 2, new paragraph added, Lines 42 to Line 51:
"In the two-dimensional analysis of Viñas and Goldstein 1991, the authors concluded the decrease of the oblique-decay growth rates into narrow band-profiles (in the wavenumber domain) when moderate oblique propagation angles of the daughter waves are considered. In the actual approach, we explore in more details the growth rates at beta values much smaller than the typical value ($\beta$=0.5) used in the previous study. The obtained solutions of the dispersion equations represented in a parallel- and    30
perpendicular- wavenumber diagram reveal new spectral features of the decay instability in low beta plasmas which have never been pointed out in former analytical investigations. Furthermore, this result strengthens the idea of multi-channel coupling of decay instability as a proof-of-concept proposed by Comisel et al. 2019 in a former study based on numerical 3-D hybrid simulations. By following the Hall MHD formalism of Viñas and Goldstein 1991, we constructed the $k_\parallel - k_\perp$ spectrum, and the results are in good agreement between the numerical simulation and the analytic treatment."                    35

*Reviewer:*

   *\* MAJOR. Discussion section is missing. Currently the manuscript is not clearly demonstrating that the obtained results are important for something. In the abstract there is a statement "Growth-rate maps serve as a useful tool for predictions of*    40

*the wavevector spectrum of density or magnetic field fluctuations in various scenarios for the wave-wave coupling processes developing at different stages in space plasma turbulence." It would be very useful if authors could discuss at least some scenario showing that their quantitative/qualitative results are applicable and important. Also it would be good to discuss the limitations of the method.*

Reply:

[revised manuscript text omitted]

*Reviewer:*

*\* lines 48-51. The description of k+- and w+- should clearly specify that those describe daughter waves. ko is vector but described in text as field-aligned wavevector, this is confusing.*

Reply:

The following comment is added in the text.

Changes in manuscript:
Page 3, Line 60 to Line 62:
', where $\boldsymbol{k}^{\pm}$ and $\omega^{\pm}$ describe the wavevectors and frequencies of the daughter waves, respectively, $\boldsymbol{k}_0 = k_0 \boldsymbol{e}_{\parallel}$ with $k_0$ - wavenumber of the Alfvén pump wave, $\boldsymbol{e}_{\parallel}$ is the unity vector parallel to the mean magnetic field, and $\omega_0$ - frequency of the Alfvén pump wave.'

*Reviewer:*

   * *line 48. What is "state vector", it is never used in the manuscript.*

Reply:
The state vectors indeed are not provided or discussed in the manuscript.

Changes in manuscript:
Page 3, Line 58:
'Each linear mode is specified by its frequency and wavevector.'

*Reviewer:*

   * *line 53. It should be more clearly explained what authors mean by "limitation for the frequency domain".*

Reply:
The sentence is rewritten at the end of the new section "Discussion".

Changes in manuscript:
Page 6, Line 159 to Line 162:
'The electron-inertia effect is neglected; one may expect that time scales or length of interest are larger than the electron gyroperiod or electron inertial length. The validity of the MHD approach with included Hall term is thus limited up to higher frequencies close to the ion gyro-frequency.'

*Reviewer:*

   * *line 87. The meaning of "domain (1,3) and (0,3)" not explained.*

Reply:
Solving the dispersion equation to provide solutions in the $(k_{\parallel}, k_{\perp})$ domain is a demanding numerical task and therefore solutions are searched in a limited wavenumber range where decay instability is expected. (for instance, the domain below $k_{\parallel}/k_0 = 1$ where modulational instability could be operational is omitted).

Changes in manuscript:
Page 4, Line 97 - Line 99:
'Because this procedure requires a demanding numerical task, the solutions of dispersion equations are searched in a limited wavenumber range where decay instability is expected (for instance, the domain below $k_\parallel/k_0 = 1$ where modulational instability could be operational is omitted).'

*Reviewer:*

    * *line 104-105. Needs more clarification how one distinguishes Alfvén and sound daughter modes, as the figure shows only the magnetic field magnitude oscillations. For parallel propagation waves are not compressional so it is surprising to see magnetic field compressions.*

Reply:
Indeed, the explanation of Fig. 3 right panel obtained from the hybrid simulation is not enough elaborated.

Changes in manuscript:
Page 4, Line 115 to Line 118:
'Magnetic field fluctuations are represented at negative wavenumbers corresponding to the backward propagating Alfvén daughter waves while the compressional forward propagating waves (acoustic-like) are shown as density fluctuations along the positive wavenumbers.'

*Reviewer:*

    * *Figure 2: caption does not specify the difference between top left and top right panels.*

Reply:
Figure 2 caption is modified as below.

Changes in manuscript:
Figure 2 caption:
' Map of the growth rates in the $k_\parallel - k_\perp$ wavenumber domain for right- (panel left) and left- (panel right) handed polarization of the Alfvén pump wave corresponding to the analysis shown in Fig. 1 at plasma $\beta$=0.02.'

*Reviewer:*

    * *Figure 3: It would be good to mark the location of pump waves booth in analytical and simulation results.*

    *right panel) are magnetic field and density fluctuations normalised to something, why are magnetic field and density fluctuations the same? It is not the case in other simulations.*

Reply:
Done. The pump wave is marked now in both panels.

Indeed, the magnetic field and density fluctuations look similar although not the same. The answer is that the 2-D wavenumber spectrum is filtered in frequency domain such that only the frequencies expected for the Alfvén daughter waves at $\omega = \omega_0 - \omega_s$ and density mode at $\omega = \omega_s$ (here $\omega_s$ is the frequency of the ion acoustic wave) are shown .

Changes in manuscript:
Page 4, Line 118 to Line 120:
'Furthermore, the 2D wavenumber spectra are filtered in frequency domain such that only the frequencies expected for the Alfvén daughter waves at $\omega = \omega_0 - \omega_s$ and for density mode at $\omega = \omega_s$ (here $\omega_s$ is the frequency of the ion acoustic wave) are shown.'

*Reviewer:*

* Outlook section should be named "Summary" or "Summary and outlook".

Reply:
Done.